# Targeting PIM Kinases to Improve the Efficacy of Immunotherapy

**DOI:** 10.3390/cells11223700

**Published:** 2022-11-21

**Authors:** Amber N. Clements, Noel A. Warfel

**Affiliations:** 1Cancer Biology Graduate Program, University of Arizona, Tucson, AZ 85724, USA; 2Department of Cellular and Molecular Medicine, University of Arizona, Tucson, AZ 85724, USA

**Keywords:** PIM kinase, immunotherapy, inflammation

## Abstract

The Proviral Integration site for Moloney murine leukemia virus (PIM) kinases is a family of serine/threonine kinases that regulates numerous signaling networks that promote cell growth, proliferation, and survival. PIM kinases are commonly upregulated in both solid tumors and hematological malignancies. Recent studies have demonstrated that PIM facilitates immune evasion in cancer by promoting an immunosuppressive tumor microenvironment that suppresses the innate anti-tumor response. The role of PIM in immune evasion has sparked interest in examining the effect of PIM inhibition in combination with immunotherapy. This review focuses on the role of PIM kinases in regulating immune cell populations, how PIM modulates the immune tumor microenvironment to promote immune evasion, and how PIM inhibitors may be used to enhance the efficacy of immunotherapy.

## 1. Introduction

The Proviral Integration site for the Moloney murine leukemia virus (PIM) kinase family consists of three evolutionary conserved serine/threonine kinases (PIM1, PIM2, and PIM3) that regulate various cellular processes, including cell proliferation, growth, motility, survival, and metabolism [1]. PIM kinases are expressed in all cell types, although tissue specificity has been identified among the isoforms. PIM1 is highly expressed in hematopoietic cells and prostate tumors, PIM2 has increased expression in lymphoid and brain tissues, and PIM3 is elevated in breast, liver, kidney, and brain tissues [2,3]. PIM kinases lack regulatory domains and are constitutively active upon translation [4]. Therefore, PIM levels are largely dictated by the rate of transcription and degradation and directly correlate with kinase activity.

PIM kinases are transcriptionally regulated primarily through JAK/STAT signaling, which is induced by cytokines, including various growth factors, interferons, and interleukins [5,6]. PIM also regulates cytokine-induced JAK/STAT signaling in a negative feed-back loop by phosphorylating and stabilizing SOCS-1 (suppressor of cytokine signaling 1). Interestingly, Pim1^−/−^ Pim2^−/−^ mice display decreased expression of Socs-1 [7]. PIM ex-pression can also be transcriptionally activated by T cell receptor (TCR) and nuclear factor kappa B (NFκB) signaling [8,9]. Less is known about the regulation of PIM protein degradation. Recent work indicates that PIM1 turnover is acutely controlled by ubiquitin-mediated degradation. For example, hypoxia, or low oxygen, has been shown to increase PIM1 protein levels by increasing its interaction with the deubiquitinase USP28, which impairs its ubiquitin-mediated degradation [10]. Notably, sustained expression of PIM1 is critical for tumor cell survival in hypoxic conditions, as PIM inhibitors selectively kill hypoxic cancer cells [11].

PIM kinases were originally identified as drivers of hematological malignancies [12,13] but have since been shown to play an equally important oncogenic role in solid tumors [14,15,16]. PIM overexpression is associated with poor survival in several types of cancer and represents a promising target for therapy [17,18]. Preclinical studies demonstrate that PIM promotes resistance to various forms of therapy, including chemotherapy [19], radiotherapy [20], and anti-angiogenic therapies [21]. As a result, PIM inhibitors have shown synergistic anti-tumor effects in combination with chemotherapy and various targeted agents. Targeting PIM to overcome resistance has previously been reviewed in detail [22]. Due to the preclinical success of PIM inhibitors, multiple small-molecule PIM inhibitors have been developed and entered clinical trials for the treatment of several cancer types, including B-cell lymphoma, acute myeloid leukemia, multiple myeloma, Hodgkin lymphoma, and prostate cancer [23]. However, PIM inhibitors have shown limited success in clinical trials to date, primarily due to toxicity. Newer generations of PIM inhibitors are more tolerated, but further work is needed to develop new and more effective PIM inhibitors. A majority of PIM inhibitors that have been developed target all three isoforms of PIM (pan inhibitors), although certain inhibitors have shown specificity for specific isoforms (Table 1). The structural and pharmacological properties of PIM inhibitors have been previously reviewed [23,24].

In recent years, it has become evident that PIM kinases modulate immune cell activity, proliferation, and survival; multiple studies have identified PIM kinases as master regulators of inflammation and immune evasion. The use of immunotherapy has shown great promise for the treatment of cancer, and there is growing interest in identifying and defining new, druggable signaling pathways that can promote immune evasion. Here, we provide an overview of what is known about the role of PIM kinases in regulating immune cell populations and examine the potential utility of targeting PIM to improve immunotherapy.

## 2. Anti-Tumor Immune Responses

The immune system is divided into innate and adaptive immune responses that work together to distinguish between self and non-self. Innate immunity activates a rapid, non-specific immune response, while adaptive immunity is specific and develops over time. Components of innate immunity include physical barriers, such as skin and mucosal membranes, as well as immune cells including dendritic cells, neutrophils, and macrophages. Adaptive immune responses rely on activation of lymphocytes, including B cells and T cells. Helper T cells (CD4^+^) are activated by recognition of peptide fragments presented in major histocompatibility complex (MHC) molecules on antigen-presenting cells, including dendritic cells and macrophages. Cytotoxic T cells recognize non-self-antigens expressed on cells, leading to direct cell killing. B cells inhibit tumor development through the secretion of tumor-reactive antibodies. Anti-tumor antibodies promote killing tumors via several mechanisms, including antibody-mediated phagocytosis by macrophages and antibody-dependent cell-mediated cytotoxicity by natural killer cells [25]. Tumor cells are first recognized through the release of neoantigens that are captured by dendritic cells. The dendritic cells process the antigens and present them on MHC molecules to T cells, leading to T-cell priming and activation. Upon activation, the tumor-specific T cells migrate to the site of the tumor. T cells recognize tumor antigens presented in MHC molecules through engagement with the T cell receptor, leading to direct cell killing [26].

At first, immune cells are able to recognize and kill tumor cells. However, over time, cancer cells evade the immune response through many mechanisms. This concept is referred to as the immunoediting hypothesis [27]. The immunoediting hypothesis can be broken down into three main stages. First is elimination of tumor cells. During active immune surveillance, immune cells recognize malignant cells through expression of tumor-specific antigens. Innate immune cells first recognize these cells, leading to cytokine cascades that suppress tumor growth. Activation of cytotoxic T cells leads to direct tumor cell killing. The next stage of the immunoediting hypothesis is equilibrium. At this point, the immune system has failed to eliminate the tumor cells, but the tumor is not progressing. The equilibrium phase can last for years. The final stage is escape, where the immune system is unable to control tumor growth [28]. Tumors evade the immune response through several mechanisms, including the expression of immune checkpoint molecules, and the recruitment of immunosuppressive immune cell populations, including tumor-associated macrophages (TAMs), myeloid-derived suppressor cells (MDSCs), and regulatory T cells (Tregs). Tumor cells are also able to evade the immune response by downregulating MHC-antigen presentation and secreting immunosuppressive cytokines that inhibit T-cell activity. Understanding the pathways that promote immunosuppression is necessary to improve anti-tumor responses.

## 3. PIM in Immune Cells

The role of PIM in regulating immune cells has largely been investigated using PIM- knockout mice. Mice lacking all three isoforms of PIM (Pim1^−/−^, Pim2^−/−^, Pim3^−/−^), known as triple-knockout (TKO) mice, are viable but much smaller than their wild-type counterparts [3,29]. Further, several hematological changes have been observed in PIM TKO mice, such as anemia, decreased peripheral CD4^+^ T cells, B cells, and granulocytes, and impaired responses of B and T cells to growth factors [3]. 

### 3.1. B Cells

The late pre-B cells of PIM TKO mice have decreased proliferation in response to IL-7 stimulation compared to those of wild-type mice [3]. IL-7 regulates B-cell differentiation and deficiencies in IL-7 signaling impair immune cell development. PIM1, in cooperation with Myc, promotes the development of pre-B-cell lymphoma in mice [30,31]. Overexpression of Pim1 and Myc in murine pre-B cells inhibits apoptosis and promotes proliferation in the absence of IL-7, resulting in impaired differentiation to immature B cells [32]. Clinically, PIM expression in diffuse large B-cell lymphomas (DLBCLs) correlates with JAK/STAT activity, increased proliferation, and advanced disease [33]. Inhibition of PIM in DLBCL cell lines decreases transcription of NFκB and p53 target genes, as well as induces MHC class II and antigen-presentation genes [34]. B cells express CD20, a cell-surface protein that regulates B-cell differentiation and activation. PIM inhibition increases CD20 levels via downregulation of MYC in patient-derived lymphoma cells. Furthermore, combined treatment with PIM inhibitors and rituximab (an anti-CD20 monoclonal antibody) synergistically suppresses tumor growth in lymphoma xenografts [35].

### 3.2. T Cells

PIM is also required for the efficient proliferation of T cells in response to TCR activation and IL-2 stimulation [3]. T cells isolated from Pim1^−/−^ Pim2^−/−^ mice are more sensitive to rapamycin than those from wild-type mice, indicating that PIM promotes T-cell growth and survival in rapamycin-treated T cells [36]. In primed CD8^+^ T cells, the costimulatory receptor CD27 induces expression of PIM1, which promotes T-cell survival independent of mTOR and IL-2 signaling [37]. Some T-cell-driven hematological cancers, such as T-cell acute lymphoblastic leukemia (T-ALL) and T-cell acute lymphoblastic lymphoma (T-LBL), have been shown to have increased PIM1 activity, suggesting the potential of PIM1 as a therapeutic target in these cancers [38,39,40]. This increased PIM1 activity can be caused by increased JAK/STAT signaling or a TCR-PIM1 translocation [39,40]. Further, PIM inhibition in T-ALL and T-LBL preclinical models suppresses tumor cell growth and improves survival [39,41,42].

PIM also affects the differentiation of CD4^+^ helper T cells. Naïve CD4^+^ T cells differentiate into several subsets, including Th1, Th2, Th17, and regulatory T cells (Tregs). Helper T cells regulate immune responses through the secretion of various cytokines. Th1 cells secrete pro-inflammatory cytokines, including IFNγ [43]. In an allogenic bone mar-row transplantation mouse model, PIM2-deficient T cells enhanced graft-vs.-host disease by promoting Th1 differentiation and expansion [44]. T-cell differentiation into Th1 helper T cells is induced by interleukin 12 (IL-12) and interferon alpha (IFNα), whereas Th2 differentiation occurs in response to IL-4. PIM1 and PIM2 expression is increased in response to Th1-specific cytokines, but not Th2-specific cytokines, in human peripheral T cells [45]. PIM kinase knockdown in human CD4^+^ Th1-polarized cells demonstrated that PIM promotes Th1 differentiation by regulating the IFNγ/T-bet and IL-12/STAT4 signaling pathways [46]. 

### 3.3. Hematopoesis

PIM also plays a crucial role in hematopoiesis. PIM expression is induced in response to cytokine signaling, which promotes growth and differentiation of bone marrow cells. Hematopoietic stem cells from TKO mice have reduced proliferation and increased apoptosis compared to those from wild-type mice. PIM TKO bone marrow cells also have a reduced ability to repopulate in transplant mouse models [29]. In contrast, transgenic mice overexpressing human PIM1 have increased hematopoiesis [47]. In summary, PIM isoforms have marked effects on immune-cell activity, proliferation, and survival.

## 4. PIM and Inflammation

Inflammation contributes to the onset and progression of many human diseases, including cancer. PIM has been shown to regulate inflammation in multiple disease contexts. In a DSS-induced colitis mouse model, PIM1 expression was increased in colon tissue and correlated with mucosal inflammation. Furthermore, PIM inhibition diminished the inflammatory response by downregulating IFNγ (Th1) and IL-17 (Th17) cytokines and upregulating TGFβ, an anti-inflammatory cytokine produced by regulatory T cells [48]. Tregs play a key role in limiting inflammatory responses. PIM1 and PIM2 phosphorylate FOXP3, the transcription factor that regulates the function of Tregs. PIM-mediated phosphorylation of FOXP3 negatively regulates Treg immunosuppressive activity [49,50]. In a CD4^+^ T-cell-mediated model of inflammatory bowel disease, PIM inhibition decreased colon inflammation, gland loss, and mucosal thickness [8]. 

One mechanism by which PIM controls inflammatory responses is through the stabilization of NF-κB. PIM1 phosphorylates RelA/p65 at Ser276, preventing ubiquitin-mediated degradation and promoting the transcription of downstream inflammatory genes [51]. In a mouse model of acute colitis, PIM inhibition decreased NF-κB activation [48]. Inflammation also promotes the development and progression of many solid tumors. Transgenic mouse models overexpressing PIM1 or PIM2 in hormone-dependent tissues have increased inflammatory immune cell infiltration. Similarly, analysis of human breast and prostate tumors indicates that PIM1/2 overexpression correlates with inflammatory features [52,53]. These results suggest that targeting PIM may be beneficial in the treatment of inflammatory-driven diseases. 

## 5. PIM and Immune Evasion

Recent studies indicate that PIM is a key player in driving immune evasion. Cancer cells often express immunomodulatory proteins that suppress an anti-tumor response and promote tumor cell survival, such as programmed death-ligand 1 and 2 (PD-L1/2). Expression of PD-L1 in tumor cells is induced by aberrant oncogenic signaling through transcription factors, such as MYC, STATs, and NFκB. In classical Hodgkin lymphoma and primary mediastinal large B-cell lymphoma cells, PIM inhibition decreases NFκB and STAT3/5 activity, accompanied by decreased surface expression of PD-L1/2 [54,55]. Additionally, T cells co-cultured with PIM inhibitor-treated cancer cells have increased expression of activation markers, indicating the importance of PIM in promoting tumor immune escape [54]. In breast cancer cells, phosphorylation of heat-shock transcription factor 1 (HSF1) at Thr120 by PIM2 promotes its stability and activates HSF1-induced PD-L1 expression. Phosphorylation of HSF1 at Thr120 is positively correlated with PIM2 and HSF1 expression in breast tumor samples [56]. 

PIM also suppresses anti-tumor immunity through the regulation of immune cell populations within the tumor microenvironment. In mice deficient in Pim2, syngeneic tumors failed to grow. Analysis of infiltrating lymphocytes in tumors isolated from WT tumor-bearing mice displayed increased levels of Tregs as compared to PIM-2^−/−^ mice. Additionally, CD4^+^ T cells isolated from Pim2-deficient mice expressed lower levels of T cell exhaustion markers (PD-1) and increased levels of T-cell proliferation and activation markers (IFNγ, TNFα, and IL-2) than those isolated from wild-type mice [44]. Interestingly, increased expression of FasL was observed on PIM2^−/−^ CD4^+^ and CD8^+^ cells, which may contribute to enhanced tumor cell killing. PIM-deficient CD8^+^ T cells were particularly important for generating and maintaining an anti-tumor response, as depletion of these T cells results in tumor relapse [44]. PIM2 has previously been shown to negatively regulate Treg activity, contradicting findings observed in this study [50]. Further studies are needed to fully understand the role of PIM2 in regulating Treg activity. Treatment with a dual-function vector containing a PIM3-targeting shRNA and an immunostimulatory ssRNA inhibits the growth of hepatocellular carcinoma cells in vivo. The dual vector promotes an anti-tumor response through increased secretion of Th1 cytokines and increases natural killer cell activation [57]. Thus, increased expression of PIM isoforms represents a new mechanism through which tumor cells escape immune surveillance. 

## 6. Immune Checkpoint Therapy

Tumor-infiltrating immune cells modulate tumor development and progression. Cytotoxic CD8^+^ and helper CD4^+^ T cells play a critical role in eliciting an anti-tumor response. However, infiltrating T cells frequently fail to suppress tumor growth due to exhaustion or disruption in activity facilitated by an immunosuppressive tumor microenvironment (TME). T cells express immune checkpoint molecules such as cytotoxic T lymphocyte-associated molecule 4 (CTLA-4) and programmed death receptor-1 (PD-1) that suppress T cell activity. Tumor cells often evade the immune response through the upregulation of programmed cell death ligand 1 (PD-L1) which binds to PD-1 and promotes T cell exhaustion [58,59]. Other immune cells such as TAMs, MDSCs, and Tregs contribute to an immunosuppressive environment. TAMs inhibit anti-tumor responses both through direct and indirect mechanisms [60]. They directly inhibit T cells by activating immune checkpoint pathways by increasing the expression of PD-L1, secreting inhibitory cytokines, and depleting metabolites in the TME. Indirectly, TAMs secrete factors that alter the vasculature and extracellular matrix, promoting T cell exclusion [60]. Similarly, MDSCs inhibit T cell activation via many different mechanisms including depleting essential amino acids that are necessary for T cell activity and proliferation [61], expressing immune checkpoint molecules [62], and secreting immunosuppressive cytokines [63]. Tregs express CTLA-4 which binds to CD80 and CD86 on antigen presenting cells (APCs), inhibiting APC-induced activation of cytotoxic and helper T cells [64]. 

A better understanding of the immune landscape of tumors has led to the development of several forms of immunotherapy. Immune checkpoint inhibitors (ICIs) targeting PD-1, PD-L1, and CTLA-4 have shown clinical success for the treatment of several cancer types. The monoclonal antibodies block coinhibitory signaling pathways, leading to increased T cell activity, and enhanced anti-tumor responses [65]. The first immune checkpoint inhibitor to be approved was ipilimumab, an anti-CTLA4 antibody [66]. Since then, many other immune checkpoint inhibitors have been developed targeting CTLA-4 and the PD-1/PD-L1 axis. PDL-1 levels are often used as a biomarker to determine whether a patient may respond well to immune checkpoint inhibitors. Tumors with high infiltration of T cells also generally respond better ICIs. Although clinical benefits have been observed, patients can develop resistance to immune checkpoint blockade. Identifying the mechanisms behind this resistance is necessary to improve the efficacy of immune checkpoint therapy.

## 7. PIM Inhibitors in Combination with Immunotherapy

PIM kinases mediate therapeutic resistance by altering a plethora of pro-survival signals. The prominent role of PIM in oncogenic signaling and therapeutic resistance has led to the development of small-molecule PIM inhibitors. Early generations of PIM inhibitors, including SGI-1176, a non-ATP-mimetic pan-PIM kinase inhibitor, reached phase I clinical trials for the treatment of non-Hodgkin lymphoma and refractory prostate cancer. However, the trial was terminated due to cardiotoxicity. Specifically, cardiac QTc prolongation was observed in patients (NCT00848601). Other PIM inhibitors are actively being tested in clinical trials. INCB053914, an ATP-competitive PIM inhibitor, has been tested as a monotherapy for the treatment of multiple hematological malignancies, including AML, multiple myeloma, and lymphoma [67]. The results of the trial indicated preliminary safety (NCT02587598). Another ATP-competitive PIM inhibitor, AZD1208, has been tested for the treatment of AML and prostate cancer. Although generally well tolerated, monotherapy was not very effective [68]. PIM447, another pan-PIM kinase inhibitor, has been tested in a dose-escalation phase 1 clinical trial for relapsed/refractory multiple myeloma. PIM447 was well tolerated, with a disease control rate of 72.2%, a clinical benefit rate of 25.3%, and an overall response rate of 8.9%. The trial was ultimately terminated due to disease progression in 54 of the 79 patients enrolled. Analysis of bone marrow did not show a clinically significant decrease in malignant plasma cells. These results suggest that PIM447 has a cytostatic effect (NCT01456689) [69]. 

The lack of clinical efficacy of PIM inhibitors as a monotherapy in hematological as well as solid tumors suggests that PIM inhibitors may best be used in combination with other therapies, which has been supported by preclinical studies. Several clinical trials are underway testing the combination of PIM inhibitors with other therapeutics. For example, PIM447 is being tested in combination with Ruxolitinib, a JAK1/2 inhibitor, and LEE011, a selective CDK4/6 inhibitor, in patients with myelofibrosis (NCT02370706). INCB053914 is also being tested in combination with a PI3K inhibitor (NCT03688152) for the treatment of diffuse large B-cell lymphoma and in combination with pomalidomide and dexamethasone for the treatment of refractory/relapsed multiple myeloma (NCT04355039). Several recent in-depth reviews have described co-targeting PIM kinases in combination with a variety of cancer therapeutics in vitro, in vivo, and clinically [22,70,71].

The role of PIM in promoting immune escape by regulating both immune and tumor cells has piqued interest in exploring whether PIM inhibitors may increase the efficacy of immunotherapies. PIM inhibitors have been shown to improve the efficacy of many other types of therapies; however, there are only a few studies to date that have explored the combination of PIM inhibitors and immunotherapies. Adoptive T cell therapy (ACT) is a type of immunotherapy where T cells are isolated from a patient, grown ex vivo, and reinjected with the goal of enhancing the anti-tumor response. However, the anti-tumor T cells often lose effector function or survival, posing a challenge for ACT. T-cell differentiation and activity are regulated through metabolic reprogramming. Upon activation, T cells switch from oxidative phosphorylation (OXPHOS) and β-oxidation of fatty acids (FAO) to glycolysis, which regulates effector T-cell activity [72]. Dependence on glycolysis also leads to the generation of terminal effector T cells, which are short lived. In contrast, central memory T (Tcm) cells are long lived and rely on OXPHOS [72]. Inhibition of glycolytic metabolism promotes T-cell memory and durable anti-tumor responses [73]. T cells treated with AZD1208, a pan-PIM kinase inhibitor, display decreased glycolytic activity, as demonstrated by reduced glucose uptake and decreased lactate production [74]. Furthermore, PIM inhibition promotes the differentiation of T cells into Tcm cells, which have lower expression of PD-1. Inhibition of PIM in adoptive T cells prolonged the survival of mice compared to those who received ACT or PIM inhibitor alone. Further studies showed that combined treatment with ACT, PIM inhibitors, and anti-PD-1 therapy significantly decreased tumor growth and enhanced survival in a melanoma xenograft mouse model [74]. These studies suggest that using a combinatorial therapeutic approach with PIM inhibitors, ACT, and anti-PD-1 may be effective for targeting tumors that are unresponsive to monotherapy. 

Immune checkpoint blockade (ICB) has been effective for the treatment of several cancer types. However, ICB has shown limited success for a large proportion of patients. An immunosuppressive tumor microenvironment consisting of myeloid-derived suppressor cells largely contributes to immunotherapy resistance. PIM1 expression is highly upregulated in MDSCs in tumors that do not respond to anti-PD-L1 therapy [75]. In vitro studies demonstrated that Pim1^−/−^ bone-marrow-derived MDSCs and wild-type MDSCs treated with AZD1208 have decreased expression of inhibitory molecules, such as arginase 1, TGFβ1, and PD-L1. Additionally, PIM1-deficient MDSCs increase proliferation of CD8^+^ T cells to a greater extent than wild-type MDSCs. In vivo studies showed that PIM inhibition enhances the efficacy of PD-L1 therapy by targeting suppressive myeloid cells [75]. Tumor-associated macrophages also promote an immunosuppressive tumor microenvironment. PIM inhibition in combination with anti-PD-L1 treatment decreases the expression of Realmα and CD206, markers of M2 pro-tumoral macrophages [75]. 

Immune cells in the tumor microenvironment have also been shown to affect PIM expression within tumor cells. In hepatocellular carcinoma tissues, PIM2^+^ cells were enriched in regions containing high levels of macrophages and T cells. Mechanistically, PIM2 expression was induced by IL-1β, secreted by IFNγ-induced pro-inflammatory macrophages [76]. Expression of PIM2 and IL-1β is increased in HCC patients treated with anti-PD-1 therapy. Interestingly, most patients with low levels of PIM2 respond fully or partially to treatment, whereas only a few patients with high levels of PIM2 respond to therapy. In a hepatoma mouse model, combination treatment of anti-PD-1 antibodies with anti-IL-1β antibodies or the PIM inhibitor AZD1208 led to complete tumor regression [76]. Although further work is needed to fully understand the role of PIM in promoting immune privilege, these studies suggest that PIM inhibitors may be useful in combination with immune checkpoint inhibitors. These initial studies suggest that combination treatment enhances an anti-tumor response by decreasing immunosuppressive immune cells and increasing T-cell activity, as illustrated in Figure 1. PIM expression may also be a good biomarker to assess which patients may respond better to immunotherapy.

## 8. Conclusions

Immunotherapies have shown robust clinical responses. However, immunotherapy efficacy varies significantly across tumor types and individuals. Understanding why certain patients respond and others do not has become a primary focus of interest. Immune suppression in the tumor microenvironment largely contributes to immunotherapy resistance. Therefore, new therapeutic approaches to target the immunosuppressive tumor microenvironment are needed. PIM kinases play an important role in regulating cell survival, growth, and proliferation in both tumor cells and immune cell populations, and they have recently been identified as regulators of immune evasion. PIM expression in tumor and immune cells inhibits anti-tumor responses, and initial in vitro and in vivo studies suggest that PIM inhibition improves the response to immune checkpoint inhibitors. Therefore, clinical studies should examine PIM inhibitors in combination with immunotherapies. Further studies are needed to better understand how PIM promotes immune evasion across different cancer types.

## Figures and Tables

**Figure 1 cells-11-03700-f001:**
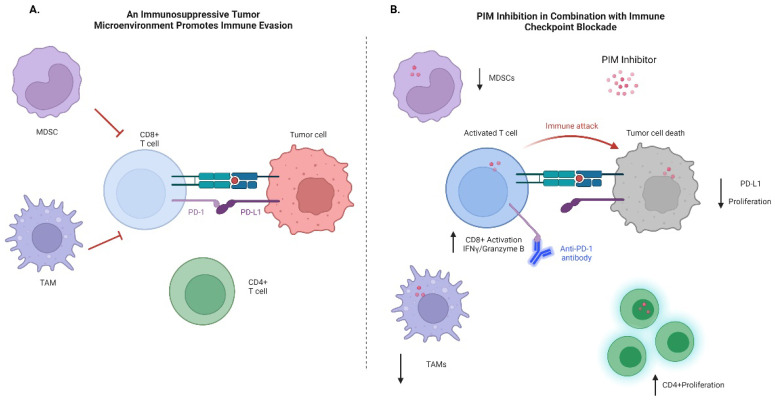
Targeting PIM kinases in combination with immune checkpoint therapy. (**A**) The tumor microenvironment contains immunosuppressive immune cells, including myeloid-derived suppressor cells (MDSCs) and tumor-associated macrophages (TAMs), that prevent anti-tumor responses. Tumor cells, MDSCs, and TAMs express programmed death ligand 1 (PD-L1), an immune checkpoint molecule that dampens cytotoxic CD8^+^ T-cell immune responses. (**B**) PIM inhibitors used in combination with immune checkpoint inhibitors decrease MDSCs and TAMs, increase CD4^+^ T cell proliferation, and increase CD8^+^ T cell activation, thereby promoting decreased tumor growth. Adapted from “Immune Checkpoint Inhibitor Against Tumor Cell”, by BioRender.com (2022). Retrieved from https://app.biorender.com/biorender-templates (accessed on 29 August 2022).

**Table 1 cells-11-03700-t001:** PIM kinase inhibitors tested clinically and experimentally.

	Isoform	Tissue Distribution	Isoform Specific Inhibitor	Pan-PIM Inhibitor
PIM Kinases	PIM1	Hematopoietic cells, gastric, head and neck, and prostate tumors	**SGI-1776** **TP-3654** *SMI-4a*	**AZD1208** **PIM447 (LGH447)** **INCB053914** **SEL24/MEN1703 *** **CXR1002** *CX-6258* *DHPCC-9* *GDC-0339* *LGB321* *AUM302 (IBL-302) **
	PIM2	Lymphoid Brain	**SGI-1776**	
	PIM3	Breast Kidney Brain	*M-110*	

**Bold**: clinically tested; *Italicized*: experimentally tested; * dual-targeted inhibitors: SEL24/MEN1703 = pan-PIM/FLT3 inhibitor, AUM302 (IBL-302) = pan-PIM/pan-PI3K/mTOR inhibitor.

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
