# Peer review of "Targeting PIM Kinases to Improve the Efficacy of Immunotherapy"

_cells, 2022, doi:10.3390/cells11223700_

Round 1

Reviewer 1 Report

The topic of the review by Clements and Warfel is timely, as there are no reviews yet focusing on the therapeutic opportunities provided by the combination of PIM inhibitors and immunotherapy. However, there are a few excellent recent reviews, which discuss also this possibility, one from the author´s own group (Toth & Warfel, 2021) and at least two from elsewhere (Luszczak et al. 2020 and Malone et al. 2020), so it would be nice to refer also to them. 

The review starts by an introduction to the basic properties of PIM kinases. As the Warfel group has published multiple original articles on them, one might expect that they are experts on this research area and familiar with PIM literature. Unfortunately the first page of the manuscript reveals that this is not the case. Even though the statements themselves are correct, the references are not.

For example, refs 2-5 are not appropriately chosen, as they do not represent the original articles, where expression patterns of PIMs were analysed. This is a very common, but unacceptable mistake that one picks pieces of information from the introductory chapters of some recent papers without digging into the original articles. Thus, these references should be replaced either by the correct ones or some reviews, where the expression patterns have been compared with each other. 

The same problem holds true for ref. 6, which is not correct, but should be replaced e.g. by Qian et al. 2005. Ref. 7 is about the role of PIMs in neuroblastoma and ref. 8 in pancreatic cancer, so why have they been chosen to tell about regulation of PIM expression by JAK/STAT signaling? And so on, so the only correct references on the first page appear to be ref. 1 to a review and ref. 12 to the author´s own work. 

The way how one refers to other people’s work is an important ethical question, so problems there raise concerns about whether or not good scientific practice has been followed during the writing. Credit should be given to those who deserve it. As already mentioned above, this should be done either by referring to original articles or to trustworthy reviews, which have referred to them. For example, Ref. 1 covers much of the contents of page 1, so in case the authors do not wish to list all the original papers, they could refer to that review plus one or two others, such as those mentioned above.

The intentional or unintentional sloppyness in the usage of references casts a dark shadow on whether also all the other refs have been chosen in an inappropriate fashion and whether or not the reader should trust the expertise of the author. Thus, to save the time and patience of the reviewers, the authors should carefully go through each and every reference to check that they are correct, before the review process can continue any further to the following pages.

Author Response

Reviewer #1 Comments

  1. refs 2-5 are not appropriately chosen, as they do not represent the original articles, where expression patterns of PIMs were analyzed. Thus, these references should be replaced either by the correct ones or some reviews, where the expression patterns have been compared with each other. The same problem holds true for ref. 6, which is not correct, but should be replaced e.g. by Qian et al. 2005. Ref. 7 is about the role of PIMs in neuroblastoma and ref. 8 in pancreatic cancer, so why have they been chosen to tell about regulation of PIM expression by JAK/STAT signaling?

We thank the reviewer for these comments.  There was an error with our Endnote formatting in the initial manuscript, thank you for bringing this to our attention.  The references have been fixed and we hope that you're opinion of our knowledge of the subject matter improves.

  1. However, there are a few excellent recent reviews, which discuss also this possibility, one from the author´s own group (Toth & Warfel, 2021) and at least two from elsewhere (Luszczak et al. 2020 and Malone et al. 2020), so it would be nice to refer also to them.

See above comments and these have been included.

Reviewer 2 Report

The authors provide an overview of the role of PIM kinases in the antitumour response and, in particular, their potential as targets to improve the efficacy of immunotherapy. The topic is interesting and well covered. However, the manuscript would benefit from some improvements.

1) In paragraph 4 (PIM and inflammation), the authors report that PIM1 and PIM2 play a negative role on the immunosuppressive function of TReg cells through phosphorylation of FOXP3. They would thus have an immunostimulatory role through the suppression of a regulatory population. This is in contrast to what is discussed below, in particular in section 6 (PIM and immunoevasion). The authors should explain this inconsistency.

2) Section 5 (Immune checkpoint therapy) is not adequately connected with the previous and subsequent parts of the manuscript. The manuscript would perhaps benefit from a reorganisation to include a first part devoted to the role of PIM in the immune response, as already done with paragraph 3, and which could also include paragraphs 4 and 6. Paragraph 5, together with 7, could be part of a separate section devoted to combination with therapies. 

Author Response

Reviewer #2 Comments

  1. In paragraph 4 (PIM and inflammation), the authors report that PIM1 and PIM2 play a negative role on the immunosuppressive function of TReg cells through phosphorylation of FOXP3. They would thus have an immunostimulatory role through the suppression of a regulatory population. This is in contrast to what is discussed below, in particular in section 6 (PIM and immunoevasion). The authors should explain this inconsistency.

Excellent point, thank you for raising this.  The role of PIM in regulatory T cells appears to be rather ambiguous and may depend context specific. We have acknowledged this inconsistency in paragraph 5 (lines 222-224).

2. Section 5 (Immune checkpoint therapy) is not adequately connected with the previous and subsequent parts of the manuscript. The manuscript would perhaps benefit from a reorganisation to include a first part devoted to the role of PIM in the immune response, as already done with paragraph 3, and which could also include paragraphs 4 and 6. Paragraph 5, together with 7, could be part of a separate section devoted to combination with therapies.

We agree that the flow of the article would be improved with these changes, so we have moved the sections on PIM and immune evasion to precede the section combination therapy.

Reviewer 3 Report

1.  Shorten the description of basic immunologic mechanisms.

2.  The authors provide a review to justify the clinical introduction of PIM inhibitors into practice.  However they do not address any barriers to such a goal.  The fact is that PIM inhibitors have been developed and introduced into clinical trials for several years.  Yet none have progressed to positive trials that lead to FDA approvals for any indication.  A "reality check" would greatly help this paper.  This could be provided by engaging a clinician-scientist with experience in kinase inhibitors to provide some commentary on the barriers that have limited PIM inhibitors thus far.  Indeed such people with PIM experience exist within the authors' institution.  Practical suggests  of ways to improve on the current inhibitors are contained within some of the quoted references.

3.  A table showing a comparison of the 3 PIM kinases (and possibly their isoforms) for their biologic properties, tissue/cell distribution, and affinity for existing and experimental PIM inhibitors would be helpful.

4.  lines 132-134.  "Pim inhibition . . . suppresses their growth and survival"  Perhaps this should be ". . . suppresses their growth and improves survival."  Suppression of tumor growth optimally would lead to improvement in the animal health and longevity.

Author Response

Reviewer #3 Comments

  1. Shorten the description of basic immunologic mechanisms.

We appreciate this comment, and understand that this section is a review of basic concepts. We made minor changes, but think that the general audience would benefit from this primer before going deeper into how PIM functions in this system.

2. The authors provide a review to justify the clinical introduction of PIM inhibitors into practice. However they do not address any barriers to such a goal.  The fact is that PIM inhibitors have been developed and introduced into clinical trials for several years.  Yet none have progressed to positive trials that lead to FDA approvals for any indication.  A "reality check" would greatly help this paper.  This could be provided by engaging a clinician-scientist with experience in kinase inhibitors to provide some commentary on the barriers that have limited PIM inhibitors thus far.  Indeed such people with PIM experience exist within the authors' institution.  Practical suggests  of ways to improve on the current inhibitors are contained within some of the quoted references.

Thank you for this comment, we added a paragraph describing the current state of PIM inhibitors in the clinic that discusses some of the reasons why PIM inhibitors have not been effective to date (Section 7, first paragraph). While clinical studies have not been overly successful,  preclinical studies have shown that PIM inhibitors may best be used in combination with other therapies as opposed to monotherapy. 

3. A table showing a comparison of the 3 PIM kinases (and possibly their isoforms) for their biologic properties, tissue/cell distribution, and affinity for existing and experimental PIM inhibitors would be helpful.

Thank you for this comment. We have included a table covering this information.

4. lines 132-134.  "Pim inhibition . . . suppresses their growth and survival"  Perhaps this should be ". . . suppresses their growth and improves survival."  Suppression of tumor growth optimally would lead to improvement in the animal health and longevity.

Thank you for this correction, it has been made in the text.

Round 2

Reviewer 1 Report

The use of references in the revised manuscript looks much more appropriate now, prompting me to read also the rest of the text, not just the first page.  

I have a few, mostly minor comments to further improve the manuscript:

Line 12 onward: When talking about PIM kinases, one can refer to either PIMs in plural or to individual family members (PIM1, PIM2, PIM3), but one should not use just the term PIM alone. However, PIM inhibitors, PIM-targeted therapy etc. are fine. So please go carefully through the text to correct PIM to PIMs, when needed.

Line 33: Cytokines include interferons and interleukins, but not growth factors, which are a different group of factors. 

Line 59: Note that toxicity has been a problem for the first generation of PIM-inhibitory compounds like SGI-1776, but since then the lack of efficacy as monotherapy has been the major problem. 

Line 66. The layout of Table 1 could be improved.

Line 70: Move the text on the efficacy of PIM inhibitors as monotherapy or in combinations here from lines 309-336. Also check the organization of chapters, so that they flow smoothly and that there are not too many repeats of things.

Line 92: … promote killing tumor kills … ?

Line 102: According to the hypothesis, the immunoediting process can be broken into… 

Line 108: The hypothesis is not in equilibrium, but the process.

Line 195: In the mouse model of acute colitis … (this model discussed already previously)

Line 201-204: I agree with these comments of another reviewer. Now the beaf after the lengthy introduction remains quite thin, reducing the impact of the review as compared e.g. to Refs 22, 70 and 71. 

Line 241: … promote tumor cell survival, such as expression of PD-L1/2.

Lines 240-270 and 272-303 identical repeats.

Line 255: Provide Ref (44?) already here.

Line 317: safety preliminarily confirmed there or how?